# Isolation of Intact Chloroplast for Sequencing Plastid Genomes of Five *Festuca* Species

**DOI:** 10.3390/plants8120606

**Published:** 2019-12-14

**Authors:** Md. Shofiqul Islam, Gretta L. Buttelmann, Konstantin Chekhovskiy, Taegun Kwon, Malay C. Saha

**Affiliations:** 1Grass Genomics, Noble Research Institute, LLC, 2510 Sam Noble Parkway, Ardmore, OK 73401, USA; msislam@noble.org (M.S.I.); kchekhovskiy@noble.org (K.C.); 2College of Agriculture, Missouri State University, 901 S. National Ave, Springfield, MO 65897, USA; 3Genomics Core, Noble Research Institute, LLC, 2510 Sam Noble Parkway, Ardmore, OK 73401, USA; tgkwon@noble.org

**Keywords:** Chloroplast, cpDNA, *Festuca* spp., forage grass, sequencing

## Abstract

Isolation of good quality chloroplast DNA (cpDNA) is a challenge in different plant species, although several methods for isolation are known. Attempts were undertaken to isolate cpDNA from *Festuca* grass species by using available standard protocols; however, they failed due to difficulties separating intact chloroplasts from the polysaccharides, oleoresin, and contaminated nuclear DNA that are present in the crude homogenate. In this study, we present a quick and inexpensive protocol for isolating intact chloroplasts from seven grass varieties/accessions of five *Festuca* species using a single layer of 30% Percoll solution. This protocol was successful in isolating high quality cpDNA with the least amount of contamination of other DNA. We performed Illumina MiSeq paired-end sequencing (2 × 300 bp) using 200 ng of cpDNA of each variety/accession. Chloroplast genome mapping showed that 0.28%–11.37% were chloroplast reads, which covered 94%–96% of the reference plastid genomes of the closely related grass species. This improved method delivered high quality cpDNA from seven grass varieties/accessions of five *Festuca* species and could be useful for other grass species with similar genome complexity.

## 1. Introduction

Chloroplasts are cytoplasmic organelles of green plants that are the active sites of photosynthesis [1]. They also play vital roles in plant physiology and development, including synthesis of chlorophyll, carotenoids, and fatty acids. Chloroplasts have their own genome of 120–160 kb in length, with the quadripartite structure found in most of land plants [2]. The chloroplast genome harbors approximately 120–130 genes that primarily encode for proteins and RNAs for the photosynthetic systems [1]. Due to high sequence conservation, the chloroplast genome sequences are correlated with plant speciation and have become an important tool in phylogenetic studies among species [3,4]. Being maternally inherited [5], the chloroplast genome is useful for developing cytoplasmic gene pools in plant breeding, and tracking the maternal parent of interspecific hybrids. To be able to take advantage of this information, it is important that the chloroplast DNA (cpDNA) be extracted with the least possible amount of contamination from nuclear and mitochondrial DNA or substances such as polysaccharides and resins in preparation for next generation sequencing [6,7]. The contamination of other genomic DNA with cpDNA reduces the quality and efficiency of the sample multiplexing method in a single sequencing run on low-output sequencing platforms such as MiniSeq, MiSeq, NextSeq, and PacBio RS II. Therefore, the isolation of high-quality cpDNA is a pre-requisite for precision and cost-efficiency of plastome sequencing. However, a high-quality extraction and purification of chloroplasts from plants has been known to be difficult to achieve [8]. Although the basic steps of varying chloroplast extraction protocols tend to reflect each other, they have often needed to be optimized to suit the properties of different plant families [6,9]. Grasses in particular have been a challenge to extract quality cpDNA from their leaves, due to high polysaccharide and oleoresin content usually present in the leaves along with other general contaminants [6]. To counter this, the use of sucrose gradients has been tested and modified for members of the Poaceae family, such as perennial ryegrass (*Lolium perenne* L.) and tall fescue (*Festuca arundinacea* Schreb.) [10,11]. Despite this optimization, the resulting cpDNA samples still exhibited a considerable amount of contamination, as concluded from the streaking present in agarose gel [10].

Within the extraction protocols for different green plants, similar sugar gradients have been used. Out of the methods tested, the two-layer gradient consisting of different densities of Percoll has been used in purifying the intact chloroplasts pellets in conifers (30%/70%), rice (*Oryza sativa* L.) (30%/80%), barley (*Hordeum vulgare* L.) (40%/80%), and the grass tribe Festuceae (40%/80%) [8,12,13,14,15]. Percoll could remove more of the polysaccharides and oleoresin contaminants than the sucrose gradients used before; however, one of the shortcomings of the Percoll gradient method of cpDNA isolation is a low yield in certain species [15]. Therefore, to further optimize the protocol for *Festuca* grass species, we implemented a single layer of Percoll solution to purify intact chloroplasts in preparation for the cpDNA extraction. The protocol developed in this experiment was optimized for *Festuca* grass species from the existing protocols. By attempting to adapt cpDNA extraction protocol to *Festuca* grasses, we aimed to develop further progress in the study of plastid genomes and to encourage more research in the sequencing of plastid genomes for different plant species.

## 2. Results and Discussion

### 2.1. Intact Chloroplast Isolation

Intact chloroplast isolation is a critical procedure used to help purify cpDNA for its application in genetic research. In the previous studies, plant-specific chloroplast isolation methods were optimized to extract cpDNA based on a high salt buffer followed by a saline Percoll gradient [8], high salt concentration buffer [16], sucrose density gradient [6,17], liquid nitrogen-sucrose gradient method [18], and high sorbitol concentration buffer, followed by a Percoll gradient [19]. Since polysaccharide and oleoresin contamination is a major problem when isolating cpDNA from many plant species [20] including *Festuca* grass species, purifying intact chloroplasts is required to extract quality cpDNA for genome sequencing. In this study, we have optimized a common method to extract cpDNA from seven grass varieties/accessions of five *Festuca* species through a single layer of Percoll solution. The plant materials were kept in darkness for 15 h at room temperature prior to extraction. This led to a significant reduction in the polysaccharides and oleoresin concentration in the extraction buffer. This strategy has been used in other protocols, in which plant materials were stored for a longer time, ranging from 48 h to 10 days at 4 °C [8,16,17] to reduce the stored polysaccharides in the extraction buffer. We reduced the storage time at room temperature enough to reduce the stored polysaccharides for *Festuca* grass species.

The next important step of intact chloroplasts isolation was leaf homogenization to harvest the maximum number of chloroplasts from plant cells. We thus adjusted appropriate blender strokes for each individual variety/accession to increase chloroplast yield in the preparation. The major challenge of this protocol was the isolation of intact chloroplasts from the broken chloroplasts in the crude pellet using a Percoll solution. A previous study [8] reported that a lower number of intact chloroplasts and a higher number of broken chloroplasts were present in the extraction when using either the modified high salt method or sucrose gradient method confirmed via microscopy image analysis. In contrast, the two-layer Percoll density gradient method resulted in the presence of abundant intact chloroplasts [8,21]. Therefore, after rapid disruption of leaf tissues in a blender, the crude chloroplasts obtained by differential centrifugation were layered on top of a 30% Percoll solution. After centrifugation, the two types of chloroplasts were separated. The intact chloroplasts formed a green pellet at the bottom of the tubes and the broken chloroplasts formed a band in the middle of the Percoll solution (Figure 1). After harvesting the pure and intact chloroplast pellet from the bottom of the Percoll layer, two subsequent washes were performed to remove Percoll residue from the preparation. This is because the isolation of cpDNA without performing these two washes could be affected by Percoll residues. As Percoll is quite expensive, the single layer method that we developed is much simpler and inexpensive compared to the existing two-layer Percoll density gradient methods.

### 2.2. Extraction of cpDNA

As the chloroplasts are double-membrane-enclosed organelles, extraction of cpDNA is largely dependent on chloroplast lysis. In this study, we have optimized the lysis step to extract a high quantity of cpDNA. Intact chloroplasts were dissolved in a lysis buffer containing RNAse A at 65 °C for 50 min with gentle mixing. The lysis step was terminated by removing heat treatment when the suspension color changed to dark olive green. Therefore, this protocol enabled us to isolate cpDNA with a high quality and yield. The isolated cpDNA formed a distinct band with a large fragment size on agarose gel (Figure 2), which indicates a high quality of polysaccharide-free cpDNA. The cpDNA of the seven grass varieties/accessions was quantified using a Qubit^TM^ 4.0 Fluorometer assay, which was designed to bind DNA from very dilute samples. The cpDNA yield was 0.25, 6.1, 5.5, 2.7, 3.0, 3.3, and 0.64 µg for Rita, PI 289654, PI 535747, Texoma MaxQ II, Resolute, Torpedo, and Vista, respectively per 40 g starting fresh leaf tissues. In a previous study, a high quantity of *Miscanthus* cpDNA (70 µg/100 g starting material) and a lower quantity of *Lolium* cpDNA (0.3 µg/100 g starting material) were purified using modified sucrose gradient method [4].

The variation in cpDNA concentration might depend on the number of intact chloroplasts present in the preparation of different varieties/accessions. It is noteworthy to mention that this protocol allowed us to purify cpDNA without the need of ultracentrifugation, which is of limited access in some laboratories.

### 2.3. Sequencing of Plastid Genomes

The optimized protocol was applied to isolate cpDNA from seven *Festuca* varieties/accessions with different ploidy levels (Table 1). To test the efficiency of the protocol and purity of cpDNA, the isolated cpDNA of the seven *Festuca* varieties/accessions was sequenced using Illumina MiSeq sequencing technology, and produced approximately 32 million reads with an average read length of 300 bp. After Illumina adapter removal and quality filtering, the contaminated mitochondrial reads were removed by performing a reference assembly against the mitochondrial genome of perennial ryegrass [22], a close relative of the *Festuca* species. The isolated cpDNA was contaminated with 0.21%–1.48% mitochondrial DNA (Figure 3).

Vista, the octoploid species, had the highest mitochondrial DNA contamination and the tetraploid accession, PI 289654, had the least. We performed reference-guided genome assembly against the plastid genomes of closely related species to estimate the number of sequence reads mapped to the reference plastid genomes. Reference-guided read mapping showed that the sequencing output contained 0.28%–11.37% chloroplast reads (Figure 4).

Vista has the highest number of chloroplast reads mapped to the reference genomes followed by Torpedo and Rita. The number of sequence reads from any individual species mapped to all the five references were fairly consistent. In a previous study, total DNA was extracted from leaf tissues of the two *Festuca* grass species (*F. altissima*, and *F. ovina*) and sequenced using Illumina Hi-Seq2000 paired-end sequencing (2 × 200 bp) [23]. They identified varied number of chloroplast reads in *F. altissima* (0.98%) and *F. ovina* (3.65%), when mapped to multiple reference genomes. The percentage of chloroplast reads obtained in this study was much higher (up to 11.27%) than those in a previous study [23]. Thus, extraction of cpDNA has an advantage over total DNA for obtaining a high number of plastid genome sequences. In the present study, the identified chloroplasts reads covered 94%–96% of the whole plastid genome of the reference species with an average read depth 10–440 fold (Figure 5). The remaining 4%–6% of the genomes leave 7–16 gaps ranging from 0.20–2.1 kb. Though there are variations in the coverage values across the entire plastid genomes, most of the genomic regions are covered with a decent number of reads in all seven grass varieties/accessions.

## 3. Materials and Methods

### 3.1. Plant Materials

Seven grass varieties/accessions from five *Festuca* species were used to develop a common protocol for extracting high quality cpDNA in order to achieve whole plastid genome sequences in this study (Table 1). The ploidy level of these plant materials ranged from diploid (2n = 2x = 14) to octoploid (2n = 2x = 56) (Table 1). Tetraploid fescue and Atlas fescue accessions were grown from a single seed, and the other five varieties were reproduced clonally from their respective mother plants. All the plant materials were grown in 15.87 cm × 16.51 cm plastic pots in the greenhouse for sufficient leaf production for cpDNA extraction.

### 3.2. Isolation of Intact Chloroplasts

The plants were kept in dark for 15 h prior to chloroplast isolation to prevent accumulation of starch and resin in the chloroplast. All the equipment and buffers used for chloroplast isolation were kept at 4 °C before extraction, and all the steps were carried out on ice or at 4 °C. Forty grams of young green leaves, not older than eight weeks, were harvested from three plants of each variety/accession. The leaves were cut into 1–3 cm pieces with scissors and were homogenized in 200 mL of 1x ice-cold chloroplast isolation buffer (CIB) with bovine serum albumin (BSA) of the Chloroplast Isolation Kit (www.plantMedia.com) to a final concentration of 0.1% (*v*/*w*) using a household blender (Oster Classic Series 5-Speed Blender, Oster, Mexico).

The leaves were homogenized using varied amounts of blender strokes between plant species, including 60 strokes for Vista; 80 strokes for Rita, Resolute, and Torpedo; and 100 strokes for Texoma MaxQ II, PI 289654, and PI 535747. The crude homogenate was squeezed out of the pulp and filtered through two layers of cotton cloth into four chilled 50-mL conical centrifuge tubes. The tubes were then centrifuged at 200× *g* for 7 min to sediment the starch, nuclei, and other cell debris. The recovered supernatant was centrifuged at 1300× *g* for 7 min to pellet intact chloroplasts. The chloroplast pellet of each tube was then suspended in 2 mL of 0.1% (*v*/*w*) CIB buffer with BSA by pipetting up and down with disposable plastic transfer pipette (5.8 mL, Samco Scientific, Mexico). Additionally, this suspension was pooled and then transferred to a Potter-Elvehjem tissue grinder with PTFE Pestles (Kimble Chase Life Science and Research Products, LLC., Vineland, NJ, USA), which was pumped up and down until the mixture became uniform.

### 3.3. Purification of Intact Chloroplasts Using Percoll

We prepared 5 mL of 30% Percoll solution in 15 mL screw cap conical centrifuge tubes by mixing 1.5 mL Percoll in 3.5 mL of 0.1% (*v*/*w*) of 1x CIB with BSA (Chloroplast Isolation Kit, SKU:30210000-1, Plantmedia, Dublin, OH, USA). Then 2 mL of chloroplast suspension were applied on top of the Percoll solution. These tubes were then centrifuged at 1300× *g* for 20 min at room temperature. The recovered intact chloroplast pellet was transferred to a 15 mL screw cap centrifuge tube and was washed twice by resuspending in three volumes of 1x CIB and re-pelleting at 1300× *g* for 5 min at room temperature.

### 3.4. Extraction of cpDNA

The washed chloroplasts pellet was lysed by suspension in 500 µL of AP1 buffer with 5 µL of RNAse A (100 mg/mL) (DNeasy Plant Mini Kit (Cat. No. 69106), QIAGEN, Hilden, Germany) and incubated at 65 °C in a water bath for 50 min. Once the suspension turned a dark olive green color, the tubes were removed. From this point onward, cpDNA was extracted using DNeasy Plant Mini Kit (QIAGEN, Hilden, Germany) and eluted to a final volume of 70 µL according to the manufacturer’s protocol. Eight microliters of cpDNA were checked for quality by gel electrophoresis on a 1.2% agarose gel in 1x TAE buffer (Figure 2). The cpDNA concentration was measured using a 1x dsDNA HS Assay kit of Qubit^TM^ 4.0 Fluorometer (Thermo Fisher Scientific, Singapore) for each sample, to determine cpDNA yield prior to Illumina MiSeq (2 × 300 bp) paired-end sequencing.

### 3.5. Next Generation Sequencing and Estimation of Chloroplast Reads

Sequencing libraries for the seven grass varieties/accessions were prepared separately using KAPA HyperPlus kit (Cat. No. KK8511) (Kapa Biosystems, Wilmington, MA, USA) according to the manufacturer’s instruction. Briefly, 200 ng of cpDNA of each variety/accession was fragmented with the KAPA Frag Enzyme and ligated with an indexed sequencing adapter. The libraries were amplified via eight PCR cycles and pooled in an equimolar ratio. The pooled libraries were sequenced using an Illumina MiSeq (2 × 300 bp read length) sequencing system. Adapter removal and trimming of low quality reads (trim quality score limit: 0.001, trim ambiguous nucleotides: 2, remove 5′ terminal nucleotides: 10, remove 3′ terminal nucleotides: 3, and discard read length below: 100 bp) were performed using CLC Genomics workbench 12.0 software (https://www.qiagenbioinformatics.com/). The number of chloroplast reads was estimated by performing a reference-guided genome assembly against the plastid genome sequences of perennial ryegrass (GenBank accession number: AM777385.2) [10], meadow fescue (GenBank accession number: JX871941.1) [23], *Brachypodium distachyon* L. (GenBank accession number: EU325680.1) [24], Italian ryegrass (*L. multiflorum* Lam.) (GenBank accession number: JX871942.1) [23], and continental tall fescue cv. KY-31 (GenBank accession number: FJ466687.1) [11] using CLC Genomics workbench 12.0 software.

## 4. Conclusions

We present a fast, simple, and inexpensive protocol to extract cpDNA, as compared to other protocols applied to *Festuca* plastid genome sequencing. This protocol was developed by modifying the existing protocols that were found useful to isolate intact chloroplasts and quality cpDNA from *Festuca* and other grass species for plastid genome sequencing. The main steps at which this protocol was modified: (1) 15 h at room temperature storage time—to reduce synthesis of starch in the leaves, (2) 60 to 100 blender strokes at the homogenization step—to harvest the maximum number of chloroplasts in the homogenate, (3) a single layer of 30% Percoll solution—to separate intact chloroplasts from the broken chloroplasts and other cell debris, and (4) incubation at 65 °C for 50 min during the lysis step—to break down the double layers of the chloroplast membrane. Therefore, this protocol has the added benefit to isolate good quality cpDNA from plants with complex genetic and genomic constitutions.

## Figures and Tables

**Figure 1 plants-08-00606-f001:**
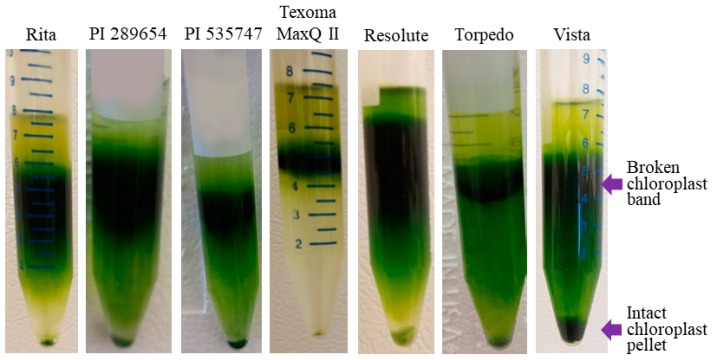
Isolating intact chloroplasts in 30% Percoll. Intact chloroplasts formed a green pellet at the bottom and the broken chloroplasts along with other plant residues formed a band in the middle of the Percoll solution.

**Figure 2 plants-08-00606-f002:**
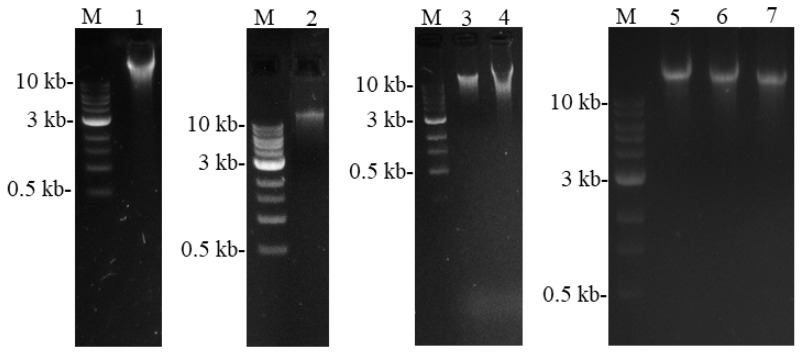
Isolated plastid DNA of seven grass varieties/accessions from five *Festuca* species. Lane M represents DNA marker; 1–Texoma MaxQ II; 2–Resolute; 3–Torpedo; 4–Rita; 5–PI 289654; 6–PI 535747; and 7–Vista.

**Figure 3 plants-08-00606-f003:**
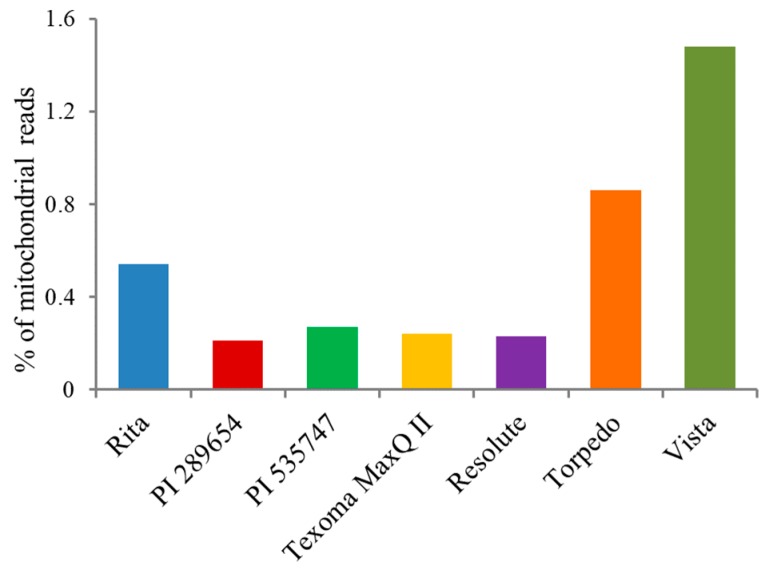
Percentage of mitochondrial sequence reads in the seven grass varieties/accessions.

**Figure 4 plants-08-00606-f004:**
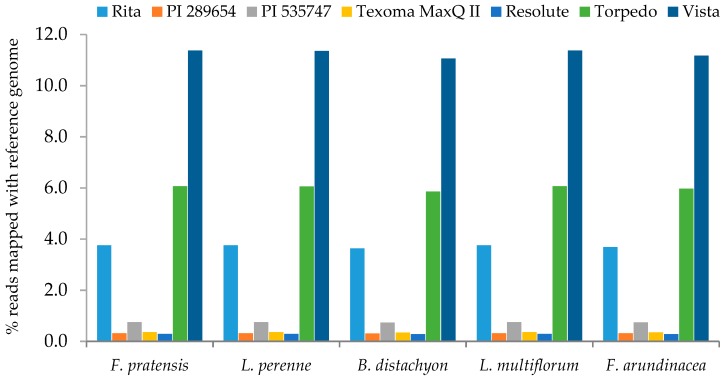
Percentage of plastid sequence reads in each variety/accession when compared to five reference plastid sequences of grass species.

**Figure 5 plants-08-00606-f005:**
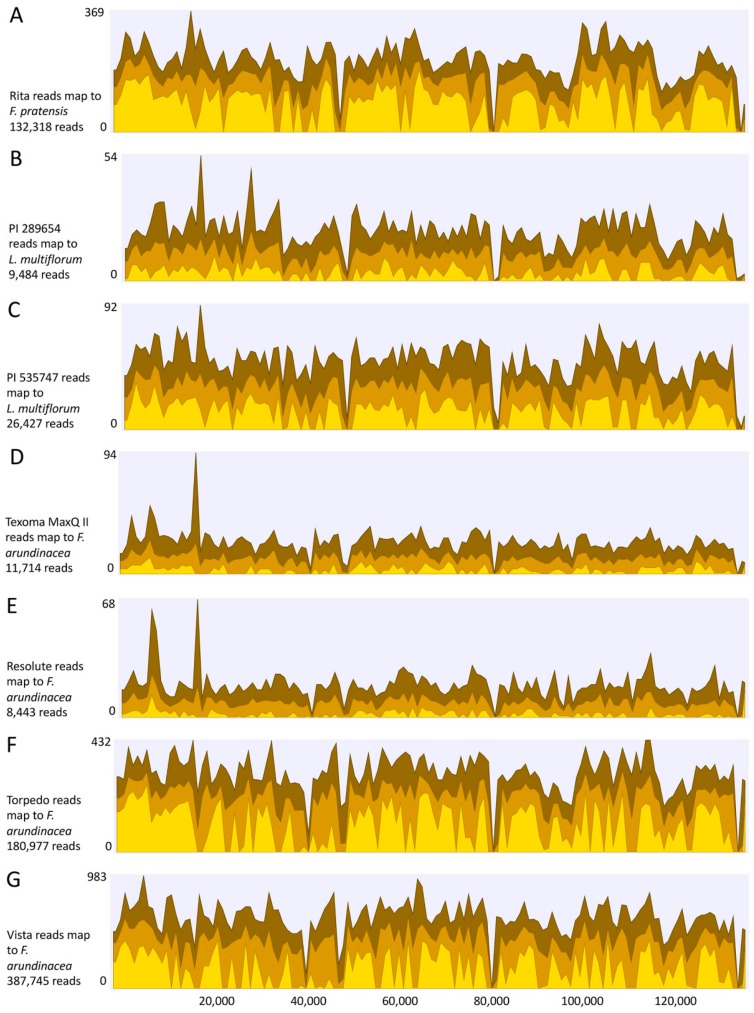
Coverage of plastid sequence reads to the reference plastid genomes. (**A**) Sequences of Rita aligned to *F. pratensis*; (**B**,**C**) Sequences of PI 289654 and PI 535747 to *L. multiflorum*; and (**D**–**G**) Sequences of Texoma MaxQ II, Resolute, Torpedo, and Vista to *F. arundinacea*. The three yellow shades represent the minimum (light yellow), average (yellow), and maximum (dark yellow) coverage values for the aggregated mapped reads (data aggregation above 100 bp).

**Table 1 plants-08-00606-t001:** List of seven grass varieties/accessions from five *Festuca* species used to develop a common protocol for cpDNA extraction.

Variety/ Accession	Common Name	Species	Type	Ploidy Status
Rita	Meadow fescue	*F. pratensis* Huds.	Bunch type	2n = 2x = 14
PI 289654	Tetraploid fescue	*F. glaucescens*	Bunch type	2n = 4x = 28
PI 535747	Atlas fescue	*F. mairei* St. Yves	Bunch type	2n = 4x = 28
Resolute	Tall fescue	*F. arundinacea* Schreb.	Mediterranean	2n = 6x = 42
Texoma MaxQ II	Tall fescue	*F. arundinacea* Schreb.	Continental	2n = 6x = 42
Torpedo	Tall fescue	*F. arundinacea* Schreb.	Rhizomatous	2n = 6x = 42
Vista	Red fescue	*F. rubra* ssp. *rubra*	Strong creeping Rhizomatous	2n = 8x = 56

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
