# Peer review of "Isolation of Intact Chloroplast for Sequencing Plastid Genomes of Five Festuca Species"

_plants, 2019, doi:10.3390/plants8120606_

Round 1

Reviewer 1 Report

Section "Introduction" should be improved.

The method of isolating chloroplasts using Percoll centrifugation has long been known, see for example Price, C. A., Cushman, J. C., Mendiola-Morgenthaler, L. R., & Reardon, E. M. (1987). Isolation of plastids in density gradients of percoll and other silica sols. Methods in Enzymology 148: 157-179. doi:10.1016/0076-6879(87)48018-x. The method was also used for monocots, cf.

Hendrik Führs, Christof Behrens, Sébastien Gallien, Dimitri Heintz, Alain Van Dorsselaer,Hans-Peter Braun, Walter J. Horst. (2010). Physiological and proteomic characterization of manganese sensitivity and tolerance in rice (Oryza sativa) in comparison with barley (Hordeum vulgare). Annals of Botany, 105(7): 1129–1140,https://doi.org/10.1093/aob/mcq046

Tanaka, M. Fujita, H. Handa, S. Murayama, M. Uemura, Y. Kawamura, T. Mitsui, S. Mikami, Y. Tozawa, T. Yoshinaga, S. Komatsu. (2004). Proteomics of the rice cell: systematic identification of the protein populations in subcellular compartments. Molecular Genetics and Genomics. 271(5): 566–576

Lehväslaiho, A. Saura, J. Lokki. (1987) Chloroplast DNA variation in the grass tribe Festuceae. Theor Appl Genet 74:298-302.

Verena Scheumann, Siegrid Schoch, Wolfhart Rüdiger.(1999) Chlorophyll b reduction during senescence of barley seedlings. Planta 209: 364-370.

It is desirable to compare diferent Percoll-based  protocols in the Introduction or Discussion..

In Ref. [2], there is not a word about optimizing the isolation of chloroplasts from different plants. But this topic is touched (in relation to the isolation of chloroplasts by centrifugation in a sucrose cushion and gradient) by Diekmann, K., et al 2008, An optimized chloroplast DNA extraction protocol for grasses (Poaceae) proves suitable for whole plastid genome sequencing and SNP detection [Ref.3]

Ref.[3] and [5] do not say about the isolation of chloroplast DNA, but the total genomic DNA.

Line 20: “cpDNA with no contamination”.  However, below, line 20 indicate that during sequencing 0.28 - 11.37% were chloroplast reads, and “The isolated cpDNA was contaminated with 0.21 –  1.48% mitochondrial DNA (lines126-27).

There is a phrase “complex plant species” in lines 225-226. Please decrypt what it means.

Author Response

Q: Section "Introduction" should be improved.

We have substantially improved the “Introduction” section. We provided additional reasoning of our research, added 10 new references to support our research and findings, and deleted two irrelevant references.

Q: The method of isolating chloroplasts using Percoll centrifugation has long been known, see for example:

Price, C. A., Cushman, J. C., Mendiola-Morgenthaler, L. R., & Reardon, E. M. (1987). Isolation of plastids in density gradients of percoll and other silica sols. Methods in Enzymology 148: 157-179. doi:10.1016/0076-6879(87)48018-x. The method was also used for monocots, cf.

Hendrik Führs, Christof Behrens, Sébastien Gallien, Dimitri Heintz, Alain Van Dorsselaer,Hans-Peter Braun, Walter J. Horst. (2010). Physiological and proteomic characterization of manganese sensitivity and tolerance in rice (Oryza sativa) in comparison with barley (Hordeum vulgare). Annals of Botany, 105(7):1129–1140,https://doi.org/10.1093/aob/mcq046

Tanaka, M. Fujita, H. Handa, S. Murayama, M. Uemura, Y. Kawamura, T. Mitsui, S. Mikami, Y. Tozawa, T. Yoshinaga, S. Komatsu. (2004). Proteomics of the rice cell: systematic identification of the protein populations in subcellular compartments. Molecular Genetics and Genomics. 271(5): 566–576

Lehväslaiho, A. Saura, J. Lokki. (1987) Chloroplast DNA variation in the grass tribe Festuceae. Theor Appl Genet 74:298-302.

Verena Scheumann, Siegrid Schoch, Wolfhart Rüdiger.(1999) Chlorophyll b reduction during senescence of barley seedlings. Planta 209: 364-370.

We reviewed all the references suggested by the reviewer and highlighted the distinctness of our method compared to the published methods.

Q: It is desirable to compare different Percoll-based protocols in the Introduction or Discussion.

We discussed different density of Percoll gradients used in the Percoll based protocols in different species and addressed the cpDNA yield obtained from these protocols (please see revised manuscript line numbers 61 – 68).

Q: In Ref. [2], there is not a word about optimizing the isolation of chloroplasts from different plants. But this topic is touched (in relation to the isolation of chloroplasts by centrifugation in a sucrose cushion and gradient) by Diekmann, K., et al 2008, An optimized chloroplast DNA extraction protocol for grasses (Poaceae) proves suitable for whole plastid genome sequencing and SNP detection [Ref.3]

It was a mistake, we cited a wrong reference. In revised manuscript we fixed it; Diekmann, K., et al 2009 has been changed to Diekmann, K., et al 2008.  

 Q: Ref.[3] and [5] do not say about the isolation of chloroplast DNA, but the total genomic DNA.

We agreed to the comments of the referee and deleted the references.

Q: Line 20: “cpDNA with no contamination”. However, below, line 20 indicate that during sequencing 0.28 -11.37% were chloroplast reads, and “The isolated cpDNA was contaminated with 0.21 – 1.48% mitochondrial DNA (lines126-27).

“cpDNA with no contamination” was revised as “cpDNA with least possible amount of contamination with other DNA”

Q: There is a phrase “complex plant species” in lines 225-226. Please decrypt what it means.

We clarified the statement.

Reviewer 2 Report

The manuscript by Md. Shofiqul Islam and colleagues reports on the determination of chloroplast genomes from Festuca species with detailed analysis of isolation of intact chloroplast DNA. Overall the manuscript is clearly written. And authors will contribute to establishment of chloroplast DNA extraction for efficient chloroplast genome assembly.

Q: In this study, authors addressed efficient CP DNA extraction method. In general, total DNA extraction methods were used for producing NGS reads. Because chloroplast DNA has unique and easily assembled for only chloroplast DNA reads. Authors should be discussion for difference of total DNA extraction and chloroplast DNA extraction and advantages.

Author Response

Q: In this study, authors addressed efficient CP DNA extraction method. In general, total DNA extraction methods were used for producing NGS reads. Because chloroplast DNA has unique and easily assembled for only chloroplast DNA reads. Authors should be discussion for difference of total DNA extraction and chloroplast DNA extraction and advantages.

We addressed the comments in the revised manuscript lines 155-161.